# An Evaluation of the NANDA International, Inc., Diagnostic Classification Among Spanish Nurses: A Cross-Sectional Study

**DOI:** 10.3390/nursrep15030079

**Published:** 2025-02-26

**Authors:** Claudio-Alberto Rodríguez-Suárez, María-Isabel Mariscal-Crespo, María-Naira Hernández-De Luis, Emília-Isabel Martins Teixeira-da-Costa, Héctor González-de la Torre, Rafaela Camacho-Bejarano

**Affiliations:** 1Research Support Unit at Maternal and Child Insular University Hospital Complex, Canary Islands Health Service (SCS), 35016 Las Palmas, Canary Islands, Spain; 2Nursing Department, Faculty of Health Sciences, University of Las Palmas de Gran Canaria (ULPGC), 35016 Las Palmas, Canary Islands, Spain; hector.gonzalez@ulpgc.es; 3Faculty of Nursing, University of Huelva, 21007 Huelva, Andalusia, Spain; mariscal@uhu.es (M.-I.M.-C.); rafaela.camacho@denf.uhu.es (R.C.-B.); 4Las Remudas Primary Health Care Centre; Canary Islands Health Service (SCS), 35213 Las Palmas, Canary Islands, Spain; nairahernandez@celp.es; 5Nursing Department, School of Health Sciences, University of Algarve, 8000 Faro, Portugal; eicosta@ualg.pt; 6Health Sciences Research Unit: Nursing (UICISA: E), Nursing School of Coimbra (ESEnfC), 3000 Coimbra, Portugal

**Keywords:** standardized nursing terminology, nursing diagnosis, surveys and questionnaires, nursing methodology research

## Abstract

**Background/Objectives**: The NANDA International, Inc., (NANDA-I) diagnostic classification is the most widely used standardized nursing language internationally. The EVALUAN-I tool was developed to evaluate the NANDA-I diagnostic classification. The aim was to analyze the use of the NANDA-I diagnostic classification among Spanish nurses and assess its correlation with sociodemographic characteristics. **Methods**: A cross-sectional study was conducted on a non-probabilistic sample of Spanish nurses working in clinical, management, and academic settings using the EVALUAN-I tool (September 2019–December 2020). The analysis was conducted using R^®^ (version 3.6.3, Lavaan package; R Core Team, 2020), with statistical significance set at *p* < 0.05. This study was approved by the Research Ethics Committee (2019-190-1). **Results:** A total of 483 responses were obtained. There was a correlation between the intensity of use of NANDA-I and its application in practice (polychoric correlation = 0.50; *p* < 0.001). Nurses with a PhD degree considered nursing diagnoses to be less evidence-based (*p* = 0.037) but more useful (*p* = 0.035). Academic and research nurses stated that NANDA-I was more useful (*p* = 0.007), even for exclusive responsibilities (*p* = 0.034), and that it provided greater significance to diagnoses (*p* = 0.0012). **Conclusions:** NANDA-I is the most widely used standardized nursing language in Spain. Nurses’ academic qualifications and work environment significantly influence their perceptions and use of NANDA-I. Advanced education fosters a critical yet positive perspective, highlighting a relationship between the intensity of its use, its application in clinical practice, and the nurse’s educational background. Tools such as EVALUAN-I promote its integration and evidence-based practice, but challenges remain in improving perceptions, scientific evidence, and visibility in electronic health records to enhance its clinical impact and nursing recognition.

## 1. Introduction

The American Nurses Association (ANA) recognizes various systems of standardized nursing languages (SNLs) that are widely used in clinical practice internationally [1]. These systems utilize both nursing-specific and multidisciplinary SNLs within electronic health records (EHRs) to enhance communication, improve the quality of care, and promote interoperability [2,3]. Additionally, they facilitate value-based healthcare and support knowledge generation [2,4]. In this sense, in the context of Australia [5], nurses identified the benefits of using SNLs, such as reducing variability in records and improving the ability to assess the effectiveness of care through outcome measurement. However, Australian nurses reported the need for improved training and education in the use of these SNLs. Additionally, there was a need to integrate these SNLs into EHRs [5]. In Europe [6], nurses face several challenges related to current documentation systems in clinical settings, limited education on SNLs, and difficulties in conducting research on SNLs. According to Dos Santos et al. [6], nurses, managers, vendors, educators, and researchers should collaborate closely to address these challenges and facilitate the implementation of SNLs in electronic documentation systems. To fully harness the benefits of SNLs, a call to action is needed to foster comprehensive collaboration between nursing practice, education, and research [6]. Additionally, in the context of Nigeria [7], nurses recognized the advantages of utilizing SNLs. However, they faced difficulties in formulating accurate nursing diagnoses (NDs) and encountered several challenges in using SNLs, including workforce shortages, lack of time, inadequate materials, and insufficient knowledge. These challenges have led to limited utilization of SNLs among nurses [7].

Among these SNLs, there are classifications of nursing interventions [8,9], nursing outcomes [10], and NDs [11]. With respect to NDs classifications, NANDA International, Inc., (NANDA-I) [1,12] is the most widely used SNL internationally [1,13].

NDs have the potential to describe and predict both the patient and organizational outcomes. However, high-quality research is needed to further explore the existence and strength of these relationships [4,14], as well as to evaluate the effectiveness of the advanced nursing process through rigorous methodologies that address clinical decision-making related to SNLs [12], particularly with regard to NANDA-I.

The use of NDs in both education and practice is increasing. However, many faculty members may be reluctant to teach these languages [15]. On the other hand, to improve nursing students’ education, the clinical use of NDs is essential. Developing nursing education curricula is crucial to help students gain a better understanding of NANDA-I’s terminology and use it effectively in clinical practice [16].

The EVALUAN-I tool was developed and validated by Spanish nurses to assess the 2015–2017 edition of NANDA-I [17] as part of a doctoral thesis conducted between 2017 and 2022 [18]. Its development included several research phases, including a qualitative phase, a descriptive phase [19], and the instrument’s development and validation phase [20]. EVALUAN-I demonstrates excellent reliability, with an excellent Cronbach’s alpha coefficient (α = 0.957) and adequate construct validity. This validity was assessed through exploratory factor analysis (EFA), which identified nine analytical dimensions: clinical competence, nurses’ reasoning skills, attitudes toward NDs, central concepts of the discipline, classification content, pathophysiological attributes, the level of scientific evidence, diagnostic precision, and conceptual correspondence between terminologies. The dimension assessing nurses’ clinical competence focused on specific NDs, which were selected after a literature review process, which is explained in a previous publication [20]. Together, these dimensions accounted for 70.86% of the explained variance. A confirmatory factor analysis (CFA) further supported the tool’s validity, with the following fit indices: Root Mean Square Error of Approximation (RMSEA) = 0.054; Non-Normed Fit Index (NNFI) = 0.903; Comparative Fit Index (CFI) = 0.910; and Tucker–Lewis Index (TLI) = 0.903 [20].

To present the results of the application of the EVALUAN-I instrument, this study aimed to analyze the use of the NANDA-I diagnostic classification among Spanish nurses and assess its correlation with sociodemographic characteristics.

## 2. Materials and Methods

### 2.1. Design

A cross-sectional observational study was carried out. This manuscript is presented in accordance with the Strengthening the Reporting of Observational Studies in Epidemiology (STROBE) Statement [21].

### 2.2. Participants and Sample

This study targeted all nurses in Spain (N = 316,094) in 2019, according to the Spanish National Institute of Statistics. A non-probabilistic convenience sampling strategy was applied across the 17 autonomous communities and two autonomous cities of Spain to ensure extensive regional representation. A sample size of 601 participants was deemed sufficient to estimate, with a 95% confidence level and a precision of ±0.04 units, the population mean of values with a common standard deviation (SD) of 0.5. An attrition rate of 0% was assumed.

The inclusion criteria encompassed nurses with an official university qualification that is recognized in Spain (Diploma of Higher Education or Bachelor’s Degree in Nursing) and who were working across various professional settings, including clinical, management, and academic roles. Nurses with less than one year of experience and those who were retired were excluded from this study. To identify participants with less than one year of professional experience, a specific question was included asking whether their professional experience was less than one year. Additionally, to identify retired participants, another question was included to determine whether they were retired or still active.

### 2.3. Variables and Instruments

The collected sociodemographic characteristics included the following variables: sex, age, level of education, work experience, professional setting, professional role, and intensity of use of SNL systems. The clinical variables were assessed using the EVALUAN-I tool [20], which comprises items that are structured into nine dimensions: clinical competence, nurses’ reasoning skills, attitudes toward NDs, central concepts of the discipline, contents of classification, pathophysiological attributes, the level of scientific evidence, diagnostic precision, and conceptual correspondence between terminologies. The attitudes section is based on an adaptation of the abbreviated version of the Position on Nursing Diagnoses (PND) Spanish version scale [22]. Responses were measured on a 6-point Likert scale, where 1 (“strongly disagree”) represents the most negative opinion and 6 (“strongly agree”) represents the most positive. The average score was 3.5 points.

The questionnaire was administered as a self-reported online instrument using the Google^®^ Forms tool. It was distributed via a web link or Quick Response (QR) code shared through email, mobile applications, and social media between September 2019 and December 2020. The questionnaire was distributed at relevant nursing conferences in Spain, including the International Research in Nursing Care meeting organized by INVESTEN and the congress of the Spanish Association of Nomenclature, Taxonomy, and Nursing Diagnoses (AENTDE). Additionally, it was made available through the AENTDE website and sent to its members via email. At the beginning of the questionnaire, participants were informed about the context and objectives of the research and were asked for their consent to participate in the study before beginning to answer the questions.

### 2.4. Data Analysis

A descriptive analysis was conducted to calculate the mean and SD of the quantitative variables and the frequency and percentage of the qualitative variables. The normality of the data was assessed using the Kolmogorov–Smirnov test with a significance level of α = 0.05, yielding a result of *p* < 0.001. Non-parametric tests were employed to compare variables between groups. Ordinal data, assumed to have an underlying continuous distribution, were analyzed using Spearman’s Rho and polychoric correlation coefficients. Polychoric correlations are particularly useful for analyzing ordinal data that are presumed to follow a continuous distribution. Unlike traditional correlations, polychoric correlation provides a more accurate estimation of the relationships between ordinal variables. This methodology is particularly valuable because it can uncover correlations that might not be evident employing other statistical tests, thereby offering greater robustness in situations where the assumptions of normality and homoscedasticity are not met [23]. A polychoric correlation coefficient closer to +1 or −1 indicates a stronger relationship between the variables, while a coefficient closer to 0 suggests a weak or no relationship. Values that are close to or above 0.5 were considered indicative of a correlation between the variables.

For comparisons involving more than two groups, the Jonckheere–Terpstra and Kruskal–Wallis tests were applied. To identify significant differences across more than two groups, post hoc analyses were conducted following the initial analysis. The Bonferroni post hoc test was applied to determine significant differences among variables with more than two groups. This test is a correction method for addressing the issue of multiple comparisons and controlling the family-wise error rate. The Bonferroni test adjusts the significance level by dividing it by the number of comparisons being made, ensuring that the risk of Type I errors (false positives) is minimized. The post hoc analyses helped clarify the pairwise group differences by identifying which specific groups were significantly different from each other [24]. Statistical significance was set at *p* < 0.05. All analyses were conducted using R^®^ (version 3.6.3, Lavaan package; R Core Team, 2020), developed by The R Foundation for Statistical Computing, Vienna, Austria.

### 2.5. Ethical Considerations

This study was approved by the Research Ethics Committee of the Province of Las Palmas (Las Palmas, Canary Islands, Spain) under registration number 2019-190-1.

## 3. Results

The results were organized into the sections below.

### 3.1. Results of Sociodemographic Analysis

A total of *n* = 483 responses were obtained from 16 autonomous communities, representing a response rate of 80.36% of the estimated sample size. The sociodemographic characteristics of the study population showed that the majority were female (*n* = 365; 75.57%), with a mean age of 43.17 years (SD = 10.03). Regarding the intensity of NANDA-I use in clinical practice, the mean score was 2.8 (SD = 1.02). The remaining variables are presented in Table 1.

### 3.2. Results of Descriptive Analysis

The EVALUAN-I scores of each dimension, along with the items with the highest and lowest values in each, are presented in Table 2.

### 3.3. Results of Inferential Analysis

The following sections outline the variables that showed statistical significance.

#### 3.3.1. Intensity of NANDA-I Utilization

No correlation was found between the variables “NANDA-I is useful” and “NANDA-I is intuitive”. However, a positive correlation was observed with “NANDA-I is applied in clinical practice” (r = 0.50; *p* < 0.001), as shown in Table 3.

#### 3.3.2. Academic Qualification

Nurses with a PhD degree perceived NANDA-I diagnoses as less evidence-based to a significant degree (*p* = 0.037). However, they considered NANDA-I to be more useful (*p* = 0.035). Additionally, PhD holders stated that both NANDA-I diagnoses (*p* = 0.03) and their related/risk factors (*p* = 0.001) were more closely aligned with medical diagnoses or diseases at a significantly higher rate than other respondents, as shown in Table 4.

#### 3.3.3. Professional Settings

Regarding the professional setting, our statistical analysis indicated significant differences between at least two groups in the perceived usefulness of NANDA-I (*p* = 0.007). The Bonferroni test revealed that academic and research nurses considered NANDA-I to be significantly more useful than those in other professional settings did.

The diagnostic label “(00126) Deficient knowledge” received high ratings across all professional settings (*p* = 0.037). The Bonferroni test indicated that nurses working in “Hospital departments” rated it higher, while those in the “Academic and research” setting rated it significantly lower than other groups. Similarly, the diagnostic label “(00030) Impaired gas exchange” received lower ratings from nurses in “Hospital departments”, particularly when compared with those working across “Several settings”.

Finally, the diagnostic label “(00095) Insomnia” showed significant differences between groups (*p* = 0.048); however, after the Bonferroni test, the differences between nurses in a “Primary care team” and those in the “Academic and research” setting were no longer statistically significant, as shown in Table 5.

#### 3.3.4. Professional Role

In terms of professional roles, statistical differences were observed in the respondents’ attitudes with a trend toward significance (*p* = 0.006). The Bonferroni analysis attributed these differences to the “Teaching and research” and “Supervisory and management” groups, highlighting the differences between academic professionals and other groups.

Additionally, significant differences were found in the responses regarding whether the use of NANDA-I was considered necessary for performing care involving exclusive responsibilities (*p* = 0.034). The Bonferroni test indicated that the “Academic and research” group differed significantly from the “Supervisory and management” group, as shown in Table 6.

## 4. Discussion

Our results indicate that NANDA-I is the most widely used diagnostic classification in Spain. Similarly, Tastan et al. [1] reported that, in an international setting, more than 72% of scientific publications relating to SNLs employed this terminology. Although the use of SNLs has the potential to enhance communication, improve the quality of care, and support interoperability, as well as facilitate knowledge generation, clinical practice often incorporates different SNLs [25]. This diversity complicates the aggregation and generalization of findings.

According to Rabelo-Silva et al. [26], the advanced nursing process involves the integration of SNLs to ensure their application in EHRs. In this regard, these authors concluded that using SNLs such as NANDA-I in electronic documentation enhances the quality of the nursing process, highlighting the need to expand the evaluation of EHRs across a greater number of international contexts. However, Santos et al. [27] emphasize the need to clarify certain epistemological aspects regarding the conceptual and operational definitions of nursing care systematization and the nursing care process, as well as their implications for professional practice perception and contribution. Additionally, healthcare facility managers play a crucial role in leading the transfer of theoretical knowledge and research findings into clinical practice [28]. In this context, nurses’ perceptions of the nursing process and its relationship with leadership are not always seen as complementary topics. Although the nursing process is recognized as a normative framework, professionals leading healthcare services are key actors in driving and promoting the systematization of nursing practice [29].

On the other hand, the applicability of NANDA-I in clinical practice is closely linked to the frequency and intensity of use, suggesting that greater professional exposure to these NDs can positively influence their adoption. Additionally, NANDA-I diagnoses have demonstrated significant clinical potential for developing assessment tools in specific situations. For instance, Martín-Dorta et al. [30,31] designed pictograms to evaluate communication abilities in patients with aphasia using NANDA-I classifications. Conversely, in certain clinical settings such as oncology care [32], nurses have shown a preference for the International Classification for Nursing Practice (ICNP) over NANDA-I due to its greater flexibility. However, these nurses also noted that none of the existing taxonomies fully address the specific contexts of oncology care when defining the severity of cancer patient’s problems. This underscores the need to integrate nursing language systems into clinical oncology contexts to improve their applicability and effectiveness [32].

The clinical applicability of NANDA-I is linked to its acceptance, as highlighted by Romero-Sánchez et al. [33]. Familiarity with NANDA-I diagnoses fosters more positive attitudes toward their use, promoting their adoption not only in clinical practice but also in academic and managerial settings [33,34]. In this regard, the findings of this study align with those of Rodríguez-Suárez et al. [34] from the Canary Islands (Spain), where a significant relationship was identified between the academic degree, work environment, and perception of the relevance of NANDA-I diagnoses. These authors also noted that similar attitudes have been observed in international research, suggesting a global trend in perceiving NANDA-I diagnoses as both a conceptual product and a practical tool.

Furthermore, attitudes toward NANDA-I diagnoses are shaped not only by external factors such as the work environment but also by the intrinsic characteristics of the NDs themselves. Significant differences in perceptions were observed based on nurses’ academic qualifications. While nurses with doctoral degrees viewed NANDA-I diagnoses as having a limited scientific basis, they still considered them valuable tools for both clinical practice and education. This apparent contrast may reflect the advanced critical thinking skills that are developed through higher education, particularly in research-oriented programs.

Similarly, Rabelo-Silva et al. [35] emphasized the importance of strengthening international collaboration in research on NANDA-I within real-world clinical settings to enhance our understanding of patient experiences and advance scientific knowledge. According to Rodríguez-Suárez et al. [13], further studies with rigorous methodologies are needed to explore clinical decision-making using NANDA-I, as well as to evaluate the effectiveness of SNLs in improving nursing interventions and patient satisfaction with the nursing process.

Nurses working in academic environments place significantly greater value on the utility of NANDA-I compared with those in clinical, supervisory, or managerial roles. However, Yalcincaya et al. [16] recommended revising the NANDA-I terminology within university curricula to enhance students’ understanding and ensure its effective application in clinical practice. According to these authors, four key aspects must be addressed: knowledge and awareness of NANDA-I, its facilitative role in patient care, the need for improved training, and the challenges associated with its use [16].

To bridge these gaps, it is essential to implement educational programs that integrate evidence-based diagnostic competencies and clinical decision-support systems [36]. These approaches not only improve diagnostic accuracy but also encourage the adoption of SNLs in healthcare, as suggested by Bertocchi et al. [37]. Strengthening diagnostic competencies through innovative educational and technological tools is particularly important for both education and clinical practice. For instance, integrating the Problem–Etiology–Signs/Symptoms (PES) format with advanced clinical decision-support systems has been shown to enhance diagnostic accuracy, an essential skill in advanced nursing practice [38].

These findings suggest that incorporating NANDA-I into education can significantly contribute to evidence-based practice, ultimately optimizing the quality of care and promoting professional autonomy. In this regard, the clinical application of NANDA-I diagnoses aligns with the establishment of an advanced nursing practice model that integrates a competency evaluation framework. This framework is based on eight key factors: evidence-based research and practice, clinical and professional leadership, mentoring and interprofessional collaboration, professional autonomy, quality management, care management, teaching, and health promotion [39]. Moreover, the integration of NANDA-I must consider the diversity of nursing education systems and cultural perspectives on clinical reasoning and professional autonomy [40]. Differences in curricula, regulatory frameworks, and the historical development of nursing as a discipline may influence the acceptance and implementation of SNLs across different healthcare contexts [41,42]. Therefore, adapting educational strategies to accommodate these variations is crucial to ensuring the effective use of NANDA-I in diverse settings.

According to Ting et al. [43], the lack of proper training and education of nurses in the use of SNLs can pose a risk to EHRs and negatively impact the quality of nursing documentation. In this context, nurses perceive EHRs as a barrier to contextualizing and synthesizing information, communicating with other professionals, and structuring patient care [44]. Therefore, ensuring early and continuous training in the use of SNLs for clinical nurses is essential [43]. In this regard, the ability to synthesize and communicate information, both individually and within teams, is a key factor in ensuring patient safety. Likewise, the design of EHRs has important implications for their proper use [44]. The alternative solutions that are employed by nurses when using EHRs highlight a significant gap in the research. Despite decades of development, the usability of these systems remains a major concern for nursing professionals. The widespread use of workarounds by the largest group of healthcare providers undermines the quality of care at all levels of the healthcare system. In this sense, further research is needed to explore the gaps in knowledge regarding these alternative practices and identify strategies to minimize their impact [45].

Finally, improvements to NANDA-I have significant implications for the global advancement of nursing, as they enhance nurses’ clinical judgment accuracy and increase their visibility within EHRs [26]. Additionally, these improvements contribute to the professional potential and scientific autonomy of nurses [46]. The clinical applicability and utility of NANDA-I are closely tied to strengthening the scientific evidence supporting its diagnoses, enhancing academic training, and increasing its clinical use in alignment with the nursing metaparadigm and core nursing principles. However, the nursing process must address the challenges arising from the complexity of this field, necessitating the conceptualization of an updated definition of the construct of human responses [47].

Regarding the limitations of this study, the self-administered nature of the EVALUAN-I instrument [20], which was distributed via social networks, may introduce selection bias and limit the generalizability of the findings due to non-probabilistic sampling. However, one of the study’s key strengths is that it represents the first application of this instrument to evaluate the use of NANDA-I in the Spanish context. Further research is needed to validate the EVALUAN-I instrument in a broader range of international settings. Additionally, future studies should examine the cost–benefit impact of implementing NANDA-I in clinical practice, as well as its influence on both organizational efficiency and patient outcomes.

## 5. Conclusions

NANDA-I is the most widely used SNL among nurses in Spain, both in clinical practice and academic training. Nurses’ academic background significantly influences their perceptions of NANDA-I. Those with advanced education adopt a critical yet constructive perspective, acknowledging its pedagogical and practical value while emphasizing the need to strengthen its scientific foundation for broader acceptance. Additionally, the work environment of nurses plays a crucial role in shaping their perceptions and usage of NANDA-I. It is more valued by professionals in academic and research settings compared with those in clinical, administrative, or supervisory roles, suggesting that the specific demands of each setting influence its adoption.

From a practical standpoint, tools such as EVALUAN-I provide valuable insights into nurses’ perspectives on NANDA-I, facilitating its integration across diverse professional contexts and promoting a more standardized, evidence-based approach. However, key challenges remain, including enhancing the perception of NANDA-I, strengthening the scientific evidence supporting certain diagnoses, and increasing its visibility in nurses’ EHRs. Addressing these challenges is essential to reinforcing NANDA-I’s impact on clinical practice and advancing the professional recognition of nursing.

While this study provides valuable insights into the integration of NANDA-I in nursing education and practice within Spain, its transferability to other contexts may be influenced by differences in nursing education, regulatory frameworks, and EHR systems. In countries where nursing education follows different competency-based models or where SNLs are not widely adopted, the impact of NANDA-I on professional autonomy and evidence-based practice may vary. Future research comparing these findings across diverse healthcare systems would be valuable in assessing the broader applicability of these results.

## Figures and Tables

**Table 1 nursrep-15-00079-t001:** Sociodemographic variables of participants (N = 483).

Variable	Statistical Results
Age; mean (SD)	43.17 (10.03)
Work experience; mean (SD)	19.89 (10.1)
Intensity of use of classification systems	
NANDA-I; mean (SD)	3.67 (0.99)
ATIC ^1^; mean (SD)	1.34 (1.06)
ICNP ^2^; mean (SD)	1.13 (0.61)
CCC ^3^; mean (SD)	1.08 (0.46)
OMAHA System; mean (SD)	0.99 (0.45)
Sex	
Female; *n* (%)	365 (75.57)
Male; *n* (%)	118 (24.43)
Level of education	
Diploma/Bachelor; *n* (%)	292 (60.45)
Master; *n* (%)	136 (28.16)
PhD; *n* (%)	55 (11.39)
Professional setting	
Hospital departments; *n* (%)	153 (31.68)
Primary care team; *n* (%)	128 (26.50)
Management; *n* (%)	48 (9.94)
Academic and research; *n* (%)	30 (6.21)
Several settings; *n* (%)	124 (25.67)
Professional role	
Clinical; *n* (%)	249 (51.55)
Teaching and research; *n* (%)	48 (9.94)
Supervisory and management; *n* (%)	55 (11.39)
Several roles; *n* (%)	131 (27.12)

^1^ ATIC: Arquitectura, Terminología, Interfase, Información, enfermerIa y Conocimiento; ^2^ ICNP: International Classification for Nursing Practice; ^3^ CCC: Clinical Care Classification.

**Table 2 nursrep-15-00079-t002:** Results for EVALUAN-I items (N = 483).

Item of EVALUAN-I	Statistical Results Mean (SD)
Clinical Competence	3.94 (1.44)
NANDA-I is useful	4.12 (1.56)
Concepts facilitate organization of specific autonomous knowledge	3.95 (1.47)
NANDA-I is essential for nursing science to progress academically	4.11 (1.66)
The structure of NANDA-I makes it easier to learn	3.65 (1.53)
The content of NANDA-I helps to decide what care to deliver	4.03 (1.47)
Using NANDA-I is compulsory to ensure that nurses are able to deliver care reflecting exclusive roles and responsibilities	3.76 (1.81)
NANDA-I is applied in clinical practice	3.53 (1.61)
NANDA-I is intuitive	3.42 (1.49)
Nursing professionals accept NANDA-I as part of their clinical practice	3.06 (1.51)
NANDA-I encourages development of nursing discipline-exclusive concepts	4.32 (1.44)
NANDA-I represents and identifies theoretical currents in nursing	3.66 (1.41)
There is internal coherence between the concepts	4.20 (1.09)
The diagnostic labels are clear and descriptive	3.87 (1.43)
The defining characteristics are relevant to diagnostic judgments	4.57 (1.23)
The configuration of the hierarchical and taxonomic structure is precise	4.14 (1.14)
Related factors or risk factors are relevant to diagnostic judgments	4.47 (1.24)
Using NANDA-I contextualizes the care to be delivered to individuals	4.08 (1.38)
Nurses’ reasoning skills	4.20 (1.57)
Diagnostic label: Risk of infection	4.79 (1.54)
Diagnostic label: Chronic pain	4.31 (1.63)
Diagnostic label: Deficient knowledge	5.18 (1.33)
Diagnostic label: Risk of perioperative positioning injury	4.73 (1.47)
Diagnostic label: Nausea	3.79 (1.67)
Diagnostic label: Impaired gas exchange	3.68 (1.64)
Diagnostic label: Activity intolerance	4.73 (1.40)
Diagnostic label: Insomnia	4.23 (1.55)
Diagnostic label: Acute confusion	3.84 (1.64)
Diagnostic label: Risk of bleeding	4.26 (1.64)
Diagnostic label: Anxiety	4.43 (1.55)
Diagnostic label: Diarrhea	3.88 (1.64)
Diagnostic label: Decreased cardiac output	3.20 (1.70)
Attitudes toward nursing diagnosis	3.84 (1.32)
Attitude: Significance	4.05 (1.30)
Attitude: Realism	3.78 (1.33)
Attitude: Obstruction	3.77 (1.30)
Attitude: Validity	3.98 (1.32)
Attitude: Relevance	3.93 (1.33)
Attitude: Gratification	3.72 (1.28)
Attitude: Creativity	3.64 (1.37)
Central concepts of the discipline	4.56 (1.29)
NANDA-I identifies the central concept of person	4.66 (1.31)
NANDA-I identifies the central concept of health	4.58 (1.25)
NANDA-I identifies the central concept of environment	4.25 (1.33)
NANDA-I identifies the central concept of nursing	4.75 (1.25)
Contents of classification	4.28 (1.33)
The content is clear and descriptive: defining characteristics	4.12 (1.33)
The content is clear and descriptive: related factors	4.21 (1.28)
The content is clear and descriptive: risk factors	4.24 (1.31)
Each concept is identified with an unambiguous code	4.55 (1.40)
Pathophysiological attributes	4.14 (1.46)
Some related/risk factors may correspond to medical diagnoses or diseases	4.26 (1.51)
Some nursing diagnoses may correspond to medical diagnoses or diseases	4.07 (1.65)
It is possible to find duplicate concepts or content in NANDA-I	4.09 (1.23)
Scientific evidence	4.44 (1.25)
NANDA-I diagnoses are based on scientific evidence	4.57 (1.25)
The design of studies offers high levels of scientific evidence	4.31 (1.24)
Diagnostic precision	3.88 (1.58)
Generally, the degree of abstraction of NANDA-I diagnoses is high	4.23 (1.30)
Diagnostic label: reflex urinary incontinence	3.69 (1.72)
Diagnostic label: sexual dysfunction	3.71 (1.71)
Conceptual correspondence between terminologies	3.92 (1.21)
Each concept has been mapped in other classifications	3.87 (1.23)
The concepts may be linked to synonymous ones in other classifications	3.96 (1.19)

**Table 3 nursrep-15-00079-t003:** Correlation with the variable intensity of the use of NANDA-I (N = 483).

Variable	Spearman’s Rho	*p*-Value	Polychoric Correlation
NANDA-I is useful	0.41	<0.001	0.45
NANDA-I is intuitive	0.26	<0.001	0.28
NANDA-I is applied in clinical practice	0.46	<0.001	0.50

**Table 4 nursrep-15-00079-t004:** Academic qualification (N = 483).

Variable	Diploma/Bachelor (*n* = 292)	Master (*n* = 136)	PhD (*n* = 55)	*p*-Value *
Evidence-based nursing diagnoses; mean (SD)	4.66 (1.22)	4.47 (1.28)	4.33 (1.33)	0.037
NANDA-I is useful; mean (SD)	4.02 (1.61)	4.10 (1.52)	4.67 (1.27)	0.035
Correspondence of NANDA-I diagnoses with medical diagnoses or diseases; mean (SD)	3.95 (1.67)	4.7 (1.52)	4.48 (1.46)	0.03
Correspondence of related/risk factors with medical diagnoses or diseases; mean (SD)	4.05 (1.58)	4.47 (1.41)	4.85 (1.09)	0.001

* Jonckheere–Terpstra test.

**Table 5 nursrep-15-00079-t005:** Professional setting (N = 483).

Variable	Primary Care (1) (*n* = 128)	Hospital Departments (2) (*n* = 153)	Management (3) (*n* = 48)	Academic and Research (4) (*n* = 30)	Several Settings (5) (*n* = 124)	*p*-Value *	Bonferroni Test	
NANDA-I is useful; mean (SD)	4.11 (1.62)	3.99 (1.53)	4.04 (1.56)	5.07 (1.03)	4.08 (1.58)	0.007	(4) vs. All groups	(1) *p* = 0.027 (2) *p* = 0.006 (3) *p* = 0.048 (5) *p* = 0.02
Deficient knowledge; mean (SD)	5.2 (1.24)	5 (1.48)	5.29 (1.3)	5.79 (0.62)	5.18 (1.34)	0.037	(2) vs. (4)	*p* = 0.03
Impaired gas exchange; mean (SD)	3.64 (1.58)	3.35 (1.72)	3.92 (1.56)	3.34 (1.8)	4.06 (1.58)	0.01	(2) vs. (5)	*p* = 0.006
Insomnia; mean (SD)	4.05 (1.56)	4.09 (1.61)	4.12 (1.68)	4.9 (1.18)	4.43 (1.43)	0.048	No significant differences	

* Kruskal–Wallis test.

**Table 6 nursrep-15-00079-t006:** Professional role (N = 483).

Variable	Clinical (1) (*n* = 249)	Teaching and Research (2) (*n* = 48)	Supervisory and Management (3) (*n* = 55)	Several Roles (4) (*n* = 131)	*p*-Value *	Bonferroni Test	
NANDA-I is useful; mean (SD)	4.06 (1.55)	4.57 (1.34)	3.71 (1.66)	4.22 (1.55)	0.033	(2) vs. (3)	*p* = 0.036
Significance; mean (SD)	4 (1.22)	4.49 (1.27)	3.71 (1.23)	4.13 (1.45)	0.006	(2) vs. (3)	*p* = 0.0012
NANDA-I is intuitive; mean (SD)	3.48 (1.47)	3.3 (1.33)	2.89 (1.52)	3.57 (1.52)	0.033	(3) vs. (4)	*p* = 0.027
Exclusive responsibilities; mean (SD)	3.81 (1.76)	4.17 (1.8)	3.2 (1.84)	3.77 (1.85)	0.044	(2) vs. (3)	*p* = 0.034

* Kruskal–Wallis test.

## Data Availability

The dataset is available on request from the authors.

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
