# Peer review of "An Evaluation of the NANDA International, Inc., Diagnostic Classification Among Spanish Nurses: A Cross-Sectional Study"

_nursrep, 2025, doi:10.3390/nursrep15030079_

Round 1

Reviewer 1 Report (Previous Reviewer 1)

Comments and Suggestions for Authors

The manuscript has benefited greatly from the comprehensive revision. Now it has become really interesting. Congratulations! 

Author Response

Reviewer 2 Report (New Reviewer)

Comments and Suggestions for Authors

The manuscript addresses a topic of great relevance for nursing practice and the standardization of nursing diagnoses. However, there are some aspects that could be improved to increase its clarity and scientific impact:

Introduction:

Add more international references regarding the use of the NANDA-I classification in contexts other than the Spanish one, to strengthen the theoretical background and give greater global scope to the study.

Methods:

It is recommended to provide more details on the methods of distribution of the online questionnaire, specifying whether a specific information campaign was carried out to reduce the risk of selection bias.

It would be useful to better clarify the inclusion and exclusion criteria, in particular specifying how the level of professional experience of the participants was verified.

Provide more details on the statistical methods used, including a more in-depth explanation of the use of polychoric correlations and post-hoc analyses.

Results:

The presentation of the results could be made clearer through the use of more synthetic tables and the reduction of repetition of information already reported.

It is appropriate to better highlight the clinical relevance of the differences found between the various professional and academic groups.

Integrate the results with a more specific discussion on how educational levels influence the perception of the NANDA-I classification.

Discussion:

Increase the comparison with international studies and critically evaluate any divergences from the results of other national contexts.

Strengthen the analysis of the practical implications for nursing education and the implementation of the NANDA-I classification in health information systems.

Language and style:

Some sentences are overly complex. They could be simplified to improve the fluency and comprehensibility of the text.

Comments on the Quality of English Language

A review of the English language by a native speaker or an experienced reviewer is suggested to refine the presentation and improve the overall fluency.

Round 2

Reviewer 2 Report (New Reviewer)

Comments and Suggestions for Authors

The manuscript presents a well-conducted cross-sectional study on the use of the NANDA-I diagnostic classification among Spanish nurses, using the validated EVALUAN-I tool. The following are some suggestions for improving the work: 1. Methods: It would be useful to add more details on the participant selection process, especially to clarify whether there were any differences between the autonomous communities in data collection. 2. Statistical Analysis: Considering the use of polychoric correlation coefficients, it could be useful to further specify the reasons for choosing this methodology, better explaining the added value compared to traditional correlations, for the benefit of less experienced readers. 3. Discussion: Some passages are repetitive with respect to the results. It is suggested to better summarize the connection between educational qualification and perception of scientific evidence of NANDA-I diagnoses, highlighting the possible cultural and educational reasons behind this difference. 4. Clinical applicability: It could be interesting to add a brief reflection on the transferability of the results outside the Spanish context, especially in countries with different nursing education and clinical documentation systems.

Comments on the Quality of English Language

.

Author Response

This manuscript is a resubmission of an earlier submission. The following is a list of the peer review reports and author responses from that submission.

Round 1

Reviewer 1 Report

Comments and Suggestions for Authors

Thank you for the opportunity to review.

The manuscript is hardly comprehensible and completely inconsistent in structure - parts of the results (sample description) can be found in the introduction and in the results, information on data analysis (category formation, statistical test procedures) (also) in the results. Approval by the ethics committee was granted in 2019, the manuscript was submitted five years later. Since the first and second study are written about, I assume "salami technique". The discussion is very broad with elements that are not addressed in the paper (e.g., cost-effectiveness of nursing interventions, advanced nursing practice, CDSS, missed nursing care).

Comprehensibility is also significantly limited because many terms are never defined or are used ambiguously (e.g., "intensity of use" - is it about the frequency of use, the spectrum of nursing diagnoses used? How does the term "application in practice" differ from this? Is it about the quality, the mode - digital or handwritten, ...? What is meant by the statements "NANDA-I is useful and NANDA-I is intuitive"? How do academic and research nurses or academic and research settings differ? "Intensity of use" is then also presented as a socio-demographic characteristic (98). Difference of management, research between setting, context, and role, also because the same terms are used differently (management and research are used as professional setting as well as professional role). Does "NANDA-I diagnoses" only include the diagnoses label if related/risk factors are listed separately (Table 3)? The discussion refers to "unhelpful", which was never explicitly surveyed. Or is "helpful" equated with "useful"?

It is not clear to me from the title, abstract, and introduction what is specifically being assessed with the tool described - "to evaluate the ... diagnostic classification", "to assess NANDA-I", "the use of NANDA-I ..." and "evaluation of nursing diagnoses". There is a significant difference between evaluating individual nursing diagnoses, the classification system (the taxonomic structure, domains/classes) or its application. In the introduction, the tool - which is used as an instrument in this paper - is described with nine analytical dimensions, under Instruments with 30 items in six sections!? If the classification system is to be evaluated, how are the clinical competence, the decision-making competence of nurses and other aspects related to it? This needs to be explained. It also remains unclear why three selected NANDA-I diagnoses (Deficient knowledge, Impaired gas exchange, Insomnia) are suddenly evaluated.  

It makes little sense to state the significance level without correlation coefficients (in the abstract).      

Formally, there are sometimes problematic sentence structures (nested sentences - e.g. 58-67) or relevant components are missing (predicate, object - e.g. sentence on software R in the abstract). Table titles are not meaningful.

Comments on the Quality of English Language

see above

Reviewer 2 Report

Comments and Suggestions for Authors

Dear authors,

I have carefully reviewed your manuscript titled " Use of the EVALUAN-I tool to assess the NANDA International (NANDA-I) diagnostic classification in Spanish nurses: Correlation analysis of a cross-sectional study " and find it to be engaging, offering valuable insights into the sociodemographic characteristics of nurses associated with the use of NANDA-I. However, I have identified some areas where improvements can be made to enhance the clarity and consistency of your paper:

Introduction: Between Table 1 and the introduction, there are discrepancies in the reported data. For instance, in the text, "Professional setting = tot. 351; Professional roles = tot. 301," whereas Table 1 shows a total of 460. Please clarify this inconsistency. I suggest including numeric data only in Table 1 and referring to it in the text (e.g., "As shown in Table 1").

Results:

·       Numeric data are redundantly presented both in the results section and in Table 1. It would be clearer to present frequency distribution data exclusively in Table 1 and reference it in the text (e.g., "As shown in Table 1").

·       Maintain consistency in the number of decimals used throughout the text (e.g., "p=0.0012" should match the format of other p-values).

·       The statement "There was no correlation between the NANDA-I is useful and NANDA-I is intuitive" requires clarification. It seems you may intend to convey, "There was no correlation between either NANDA-I intensity and NANDA-I is useful or NANDA-I intensity and NANDA-I is intuitive." Please clarify this sentence for better understanding.

·       Review the usage of "lower" and "higher" in the results section concerning the diagnostic label "(00126) Deficient knowledge." Table 1 indicates that "Academic and research" has higher values than the other groups, whereas the text states the opposite.

Discussion: Discuss the finding that "PhD degree significantly estimated that both NANDA-I diagnoses and their related/risk factors corresponded more closely to medical diagnoses." Nursing diagnoses are integral to nursing clinical judgment and decision-making, offering a holistic and person-centered perspective distinct from medical diagnoses. The literature suggests that nursing diagnoses provide additional information beyond medical diagnoses, which may influence patient outcomes.

Grammar: Ensure consistent use of the Oxford comma throughout the text (e.g., abstract). For numbers below ten, consider writing them out in words (e.g., "four," "nine," etc.).

These revisions will help improve the clarity and coherence of your manuscript, enhancing its overall impact. Thank you for considering these suggestions.

Reviewer 3 Report

Comments and Suggestions for Authors

Dear authors

thank you for this interesting paper which will encourage nurses to use and to keep working nursing diagnosis and NANDA-I. 

The paper is well written and clear. I would suggest removing table 1, as this is repeating the context of the text in results. 

Also, in the conclusion, more details are needed in how to use this in further research. Regards

Reviewer 4 Report

Comments and Suggestions for Authors

Dear authors thank you for allowing me to review this interesting manuscript presenting the secondary analysis performed in a sample of more than 400 nurses concerning the application of the NANDA taxonomy in sub-groups of participants.

Although the topic is of interest for the nursing community, I found this manuscript very poorly presented and sustained by a definition of the gap of knowledge, presentation of the rationale, and definition of hypotheses. Specifically, I found some passages very hard to follow and I have the impression that in some parts you have forgotten the use of the verb in your sentences.

I have provided some suggestions below to help you improve the overall quality of your manuscript.

Abstract:

Lines 29-30. There are two sentences on analyses and ethical approval missing the verbs. Moreover, it is not specified by what Ethical Committee the approval has been provided.

Lines 30-34 if all p-values are describing correlations (as I have expected from the title and the first part of the abstract) they should be corroborated by the correlation coefficient.

BACKGROUND

Page 2, Lines 58-80 all these are results from the previous study (ref 6) and not providing any gap of knowledge, rationale, and hypotheses. These should be the backbone of a background, and they are totally missing here.

If I mind, the objective is very vague. What do you mean by "analyse the use"?

METHODS

 Page 3, line 101. Is it a new sentence starting here?

Data analyses are not explained, the concept of using a correlation analysis (which is a weak analysis plan) is totally missing here, as those relating to the other analyses.

Results

All the first part (Page 3, Lines 120-140) are not results but data management that should be reported in the methods and not here.

Page 4, Lines 145-146. This is why I have asked to substantiate the analyses plan. I need a reference to believe your correlation coefficient is reliable, in my memories is 0.7, but this issue could be resolved explaining better in the methods and providing a reference for the interpretation of results.

Moreover, why did you performed a correlation and not a chi-square or McNemar test? You could split your sample in high and low and identify differences in the two groups rather than relying a non-specific analyses as correlation.

Table 4. The last column is not needed, you could use the * to show significant values.

Discussion is disconnected from your results, there are several parts that do not met with the data you have reported. Please, read Page 6, Lines 221-228 (very carefully lines 227-228 as they are awkward) and identify how these connect with your results.

I hope these brief suggestions will allow you to improve the quality of your manuscript.

Best regards

Comments on the Quality of English Language

Dear Authors, I am concerned that the quality of your English phrasing is very poor. Since for the abstract and throughout the text, some sentences are missing the verb(s), I have provided some examples in the comments for you. 

My suggestion is to make an extensive revision of your manuscript with the help of a professional editing service.